# Ecological Potential of Mediterranean Habitats for Black Pine (*Pinus nigra* J.F. Arnold) in Croatia

**Damir Barčić [1], Vladimir Hršak [2], Roman Rosavec [1],\* and Mario Ančić [1]**

[1]  Faculty of Forestry and Wood Technology, University of Zagreb, 10000 Zagreb, Croatia
[2]  Faculty of Science, University of Zagreb, 10000 Zagreb, Croatia
\*  Correspondence: rosavec@sumfak.hr

**Abstract:** This study outlines research performed on experimental plots established in forest plantations and in natural black pine forests. The objective of the study was to determine the habitat factors that have the greatest impact on the growth and development of natural black pine forests and black pine forests plantations upon the return of climatozonal vegetation. Plots 625 m$^2$ in size were selected and vegetation inventories made, and the structural elements of black pine plantations examined. Multivariate analysis found that elevation and slope were the two variables that had the highest positive correlation with the floristic composition of the vegetation. Furthermore, one of the goals was to determine the differences with regard to reforestation with black pine. The analysis of the isolated experimental plots showed a clear grouping of plots according to habitat factors. The differences in the synecological factors in the research area resulted in the presence of sub-Mediterranean, epi-Mediterranean and eu-Mediterranean elements within the black pine forest plantations. This also reduces the risk of the occurrence and development of large wildfires.

**Keywords:** reforestation; succession; forest vegetation; pine plantations; karst





## 1. Introduction

In terms of amelioration, the role of black pine throughout the Mediterranean karst area is exceptionally valuable. Black pine is perhaps the most important species for karst reforestation due to the various ecological and social roles it plays, in addition to its commercial importance. It also plays an irreplaceable role in preventing habitat degradation and the prevention erosion from expanding. This has contributed to its widespread distribution, from the western to the eastern Mediterranean. According to Quézel and Médail [1], black pine is represented by six subspecies in the Mediterranean region. Black pine is the most widely distributed pine species in mountain areas of the Mediterranean Basin and commonly used for afforestation [2]. These pine forests are classified as "habitats of European interest" and require specific conservation measures according to the Convention for the Conservation of European Wildlife and Natural Habitats (Resolution 4/1996), due to the lack of successful natural regeneration [3].

Black pine has been used as a reforestation species throughout virtually the entire area the northern Adriatic coast in Croatia. Particular success was achieved during the work and establishment of the Inspectorate (i.e., Inspectorate for the forestation of karst, bare soils and flood management in the northern Adriatic; [4]). During the 20th century, black pine was used as a pioneer species to stop erosion processes, degradation and devastation, not only in the Dinaric karst area, but also in the Apennine Mountains [5–7]. Though the ecological and production potential of black pine is prominent in the Dinaric karst, this sub-Mediterranean area is largely threatened with the risk of wildfires [8,9]. Nevertheless, increasing land abandonment in Mediterranean Europe during the last decades has favoured forest recovery [10,11] including *Pinus nigra* J.F. Arnold. stands [12]. Meanwhile throughout its European distribution range [13,14], black pine has important

economic value due to the high quality of the wood products, and also for its strong ecological and protective roles (i.e., black pine forests are included in the EU Habitats Directive under no. 9530*(Sub-Mediterranean pine forests). In sub-Mediterranean habitats, black pine has certain limitations, above all, periods of drought and water deficits. In the western Mediterranean, this is a key factor in the regeneration of forest ecosystems [15]. Projected decreases in precipitation and increases in temperature in future climate scenarios may aggravate their vulnerability and alter their current distribution in the Mediterranean Basin [3]. In addition to the high vulnerability of sub-Mediterranean areas to global climate change [16] and changes in the seasonal distribution of precipitation over the past decade [17], it is evident that these changes have had negative effects on forests and their stability.

Black pine was one of the fundamental reforestation species during the great forest planting project (starting 150 years ago) in Croatian forestry [18]. In Croatia, the distribution, phytocenology of black pine forests and taxonomic research have been examined by many authors [19–21]. According to the Flora Europaea [22], it is possible to differentiate two subspecies of black pine in Croatia, ssp. *nigra* and ssp. *dalmatica*. Vidaković [23] listed three taxa of black pine in Croatia: *Pinus nigra* ssp. *austriaca*, *Pinus nigra* ssp. *dalmatica* and *Pinus nigra* ssp. *gočensis* var. *illyrica*. However, according to Liber [21], Vidaković's report of the taxon ssp. *gočensis* is questionable, as this is likely just an ecotype characteristic for serpentine substrates.

The habitat conditions of the study area where the black pine stands are found in the western part of the Croatian Dinaric karst are best characterised by their heterogeneity. Erosion is also an important driver of land degradation, in that case afforestation effort in troubled areas is an expensive practice; the success of afforestation is not guaranteed, particulary in degraded soils [24]. This area is encompassed by the Mediterranean phytogeographic area, and is also under the strong influence of the Euro-Siberian–North American phytogeographic area. Moreover, the diversity of environmental conditions, geological and geomorphological factors and edaphic conditions can be associated with a higher degree of vulnerability to various erosion processes. Defining those relationships in karst, and determining the role of black pine stands in those processes, is therefore an important aspect of researching forest amelioration. A complicating circumstance for the management of these forests is the occurrence of forest fires, and subsequent regeneration after fires. Therefore, Santana [9] warns of the key issues to address after fire: when and how to extract the wood mass? The reason for this is the need to reduce all risks of erosion processes. Simultaneously, Mediterranean habitats are under very strong and prolonged drought in the recent period because plants of the Mediterranean region are exposed to drought stress [25,26].

This paper compares and analyses the possibilities of the habitat and potential of black pine stands under varying conditions. Namely, habitat conditions (slope, shallow soil, fire) largely govern the management of black pine in the Dinaric karst in Croatia. It should also be noted that this area also falls within the NATURA 2000 ecological network, where management recommendations require finding a balance between production objectives and conservation needs [27]. Therefore, one of the goals in that study suggested the need for appropriate silviculture works aimed at increasing the quality of black pine stands, and for their future management. In addition, in our study one of the objectives is to find possible differences between black pine plantations (established by reforestation) and natural pine stand. Furthermore, an analysis of the habitat conditions and production potential indicate the possible fulfilment of the ecological (transition towards mixed deciduous stands) and the economic roles (quality of wood products, biomass, etc.).

## 2. Materials and Methods

### 2.1. Study Area

Experimental plots were set up in black pine stands in western and southwestern Croatia (northern Adriatic region). Plot selection was based on data from the Forestry

Management Plan, and an overview of the geological, pedological and phytocenological maps, taking into account certain factors such as stand age, human impacts, microrelief, elevation, slope and insolation. All plots were set up in stands over 40 years old. Of the 50 experimental plots, the majority (44) were set up in forest plantations, while 6 plots were set up in natural black pine stands (Figures 1 and 2). Study area is represented by the Cfsax" climate according to Köppen. It is a moderately warm rainy climate with hot summers and an average monthly temperature above 22 °C. In the structure of the land cover, skeleton-rich soils of shallow Eutric Cambisol on carbonate rocks, marl and flysch, red soil (terra rossa) and rendzina on carbonate rocks predominate. Surface stoniness and rockiness often contribute to low surface productivity ratings.

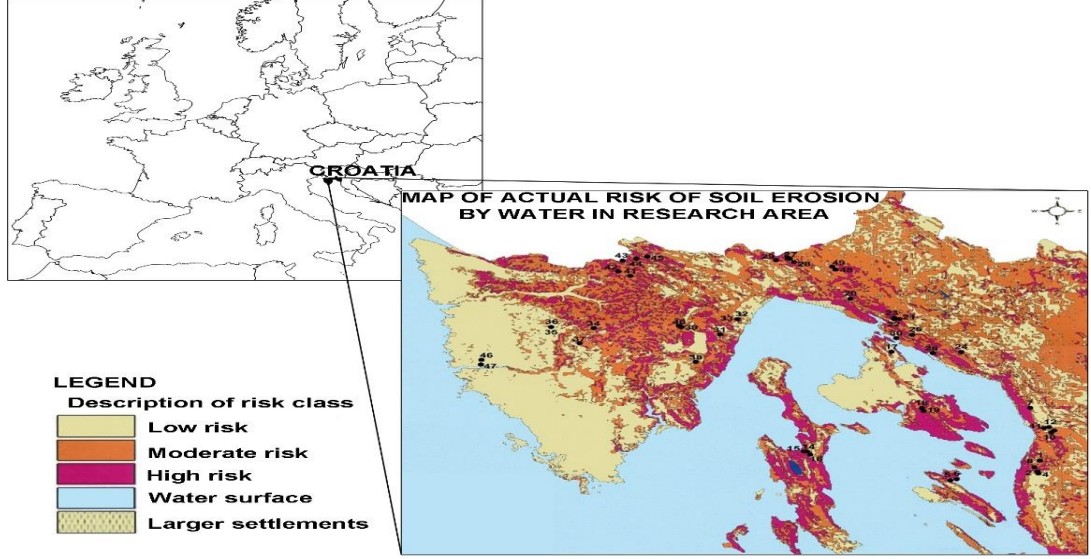

**Figure 1.** Distribution of plots in the study area in Croatia.

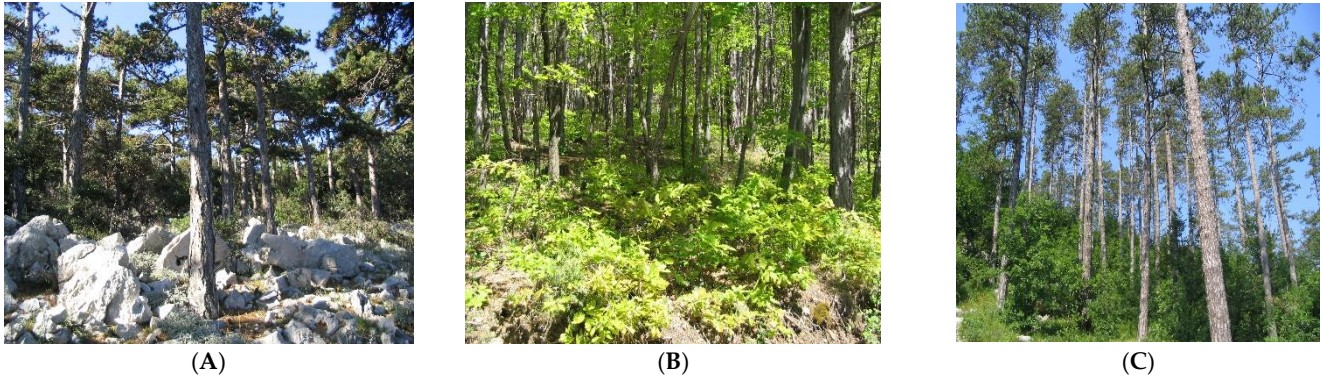

**Figure 2.** Black pine stands ((**A**)—stand on 196 m, (**B**)—stand on 539 m, (**C**)—stand on 716 m).

The bioclimatically explored area (Table 1) is humid and subhumid with moderate cold climate. The maximum precipitation is in late fall or early winter. North and northeast winds dominate.

**Table 1.** Vegetation and climate properties of the study area (Source: research and customized data of State hydrometeorological institute).

| Vegetation Belt | Vegetation Zone | Phytocenosis | Average Annual Precipitation | Average Annual Temperature | Pioneer Species | Climatogenic Species |
|---|---|---|---|---|---|---|
| Mediterranean-litoral | Sub-Mediterranean | *Querco pubescenti-Carpinetum orientalis* Horvatić 1939 | 1200 mm | 12–14 °C | *Pinus nigra* | *Carpinus orientalis, Quercus pubescens* |
| Mediterranean-montan | Epi-Mediterranean | *Aristolochio luteae-Quercetum pubescentis* (Horvat 1959) Poldini 2008 | 1400 mm | 10–12 °C | *Pinus nigra* | *Ostrya carpinifolia, Quercus pubescens* |

*2.2. Data Collection*

Stand structure analysis was performed on the experimental plots (each 625 m$^2$ in size). For each plot, black pine trees were measured by number of trees, basal area and wood volume. Wood volume was calculated according to the volume tables and yield according to the yield-tables [28]. Vegetation relevés were performed on each experimental plot using the phytosociology methodology [29,30]. The abundance and cover of species was estimated using the expanded scale according to Barkman et al. [31]. Assessments were transformed on an ordinal scale according to van der Maarel [32], prior to analysis. Plant nomenclature was determined according to Nikolić [33–35]. Field studies on experimental plots also included the sampling of the humus layer formed from five individual samples within each experimental plot. Samples were taken for the determination of soil reaction, humus and total nitrogen content, and the C/N ratio. Depending on the degree of acidity or pH value of soil, the dominant pedogenic processes in the soil can be determined. To calculate the quantity of forest floor, three samples were taken along the diagonal of a 25 × 25 cm experimental plot. The procedure was performed in accordance with the UNECO guidebook [36].

*2.3. Data Analyses*

Laboratory analyses were performed on samples of the humus layer:

1.  Soil reaction was measured electrometrically with a combined electrode in a suspension of soil in water, i.e., in 0.01 M CaCl$_2$ in the ratio of 1:2.5 for surface mineral and argyl-accumulative layers, and in the ratio of 1:10 for prominently humic surface horizons. A laboratory microprocessor pH meter (MA 5736, Metrel, accuracy ± 0.01 pH) was used for the measurements.
2.  Humus content was determined using the bichromatic method according to Tjurin.
3.  The total nitrogen content was determined by incineration according to the Kjeldajl procedure and distillation according to Bremner.
4.  Samples of the forest floor were ground and homogenised and incinerated in a microwave oven with HNO$_3$ (Milestone Laboratory System). K content was determined by flame-photometry, and all other elements determined using the AAS method (atomic absorption spectrophotometry).

2.3.1. Statistical Analysis

First, cluster analysis was performed using the Soerensen (Bray-Curtis) distance as resemblance measure with the group average as clustering method [37]. The obtained results are arbitrarily divided into 5 clusters, which are further analyzed separately. Indicator Species Analysis (ISA) was used to determine species that are relevant for particular clusters obtained by cluster analysis [38]. Indicative types were those with $p < 0.05$.

2.3.2. Multivariate Fuzzy Set Ordination

The analysis was done according to [39,40] using PCOrd 7.01 software. As this analysis is an important issue and order of environmental variables, an initial preliminary analysis

was made to determine the number and order of the variables involved in the analysis (such as slope, altitude, organic matter, total nitrogen). The analysis was first made for the entire set of data and then each cluster was analyzed individually to determine the importance of individual variables for the vegetation structure. As a resemblance measure in MSFO, the Soerensen (Bray–Curtis) distance was used.

## 3. Results

According to the cluster analysis, the obtained optimal number of cluster was five. Figure 3 showed the first cluster included 29 plots, the second 7 plots, the third 5 plots, the fourth 5 plots and fifth 4 plots. In the first cluster, favorable habitat conditions positively influenced the presence of sub-Mediterranean vegetation, while the emphasis here is also on the economic potential of certain forest cultures (P22, P27, P28, P29, P31, P32, P33). Wood volume in these plots ranged from 210.56 $m^3$/ha to 411.44 $m^3$/ha. Ecologically, this plot group fell within the sub-Mediterranean and epi-Mediterranean zone, at the border area of the montane vegetation belt that is dominated by common beech.

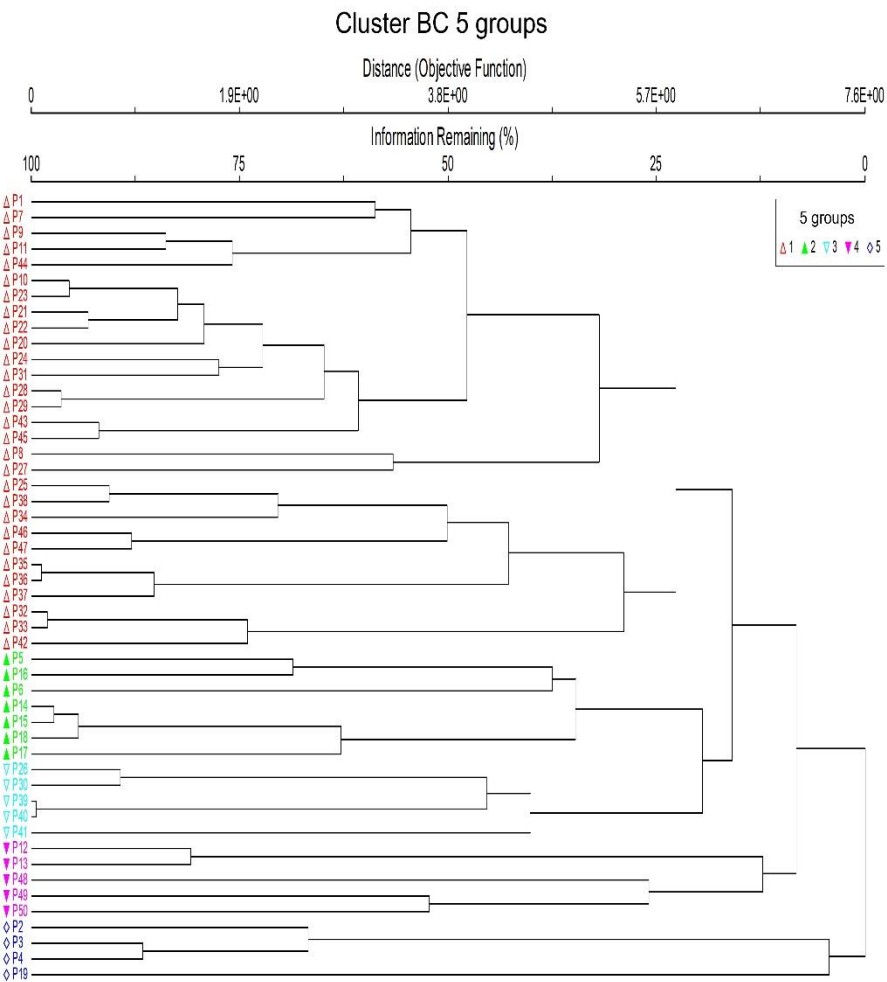

**Figure 3.** Cluster analysis of experimental plots obtained results are arbitrarily divided into five clusters.

It is significant that the second cluster included virtually all island plots (island of Rab, Cres, Krk) in the warmer part of the sub-Mediterranean zone (P5, P6, P14, P15, P16, P17, P18). According to the phytocenological relevés, this is a transition area between the eu-Mediterranean and sub-Mediterranean zones. In the economic sense, the state of the forest cultures is quite poor. Wood volume deviates from the prescribed management plans, and ranges from the minimum 53.12 $m^3$/ha to the maximum 282.88 $m^3$/ha. The third

cluster covers plots along the seashore at lower altitudes. Plots on the flysch substrate are also grouped here (P39, P40, P41).

The fourth cluster included plots in natural black pine stands (P48, P49, P50) and forest plantations in a protected area (P12 and P13). The joint characteristics of these plots is the greater slope and higher elevation in the natural black pine (*Euphorbio triflorae-Pinetum nigrae* Trinajstić 1999) stand. In the ecological sense, this is the sub-Mediterranean and epi-Mediterranean vegetation zone. In terms of economic potential, given the habitat quality class (IV), the condition is poor. The fifth cluster included three plots in a natural stand of the association *Cotoneastro-Pinetum nigrae* Ht. 1938. (P2, P3, P4), and one plot (P19) in the forest culture in the peak zone of the island of Krk. The common ecological characteristic of this cluster is the exceptionally strong impact of the dominant cold northeasterly wind (Bura). This circumstance impacts the poor economic potential of black pine in this area.

The importance of individual gradations is presented with Cum-R2 and Inc.-R2 (Table 2). These values are highest for altitude and slope. The nutrient gradient has a significantly lower value. The R2-Max value shows the association of environmental gradients with the vegetation structure. For altitudes and organic matter, it is smaller than the Cum-R2, which indicates good connectivity. The R2-Max slope gradient is larger than Cum-R2, indicating a weak connection. All this is confirmed by the biplot (Figure 4). The x-axis biplex represents the height gradient and the y-gradient represent nutritient gradient.

**Table 2.** Results of multivariate fuzzy set ordination indicate important values of altitude and slope.

| Variables | Cum-R2 | Inc.-R2 | Random *p* | Gamma | R2-Max | R2-Min |
|---|---|---|---|---|---|---|
| Altitude | 0.1973 | 0.1973 | 0.0010 | 1.0000 | 0.1911 | 0.0001 |
| Slope | 0.3210 | 0.1072 | 0.0690 | 0.8972 | 0.3411 | 0.0023 |
| Org mat | 0.3922 | 0.0712 | 0.1960 | 0.2251 | 0.3685 | 0.0186 |

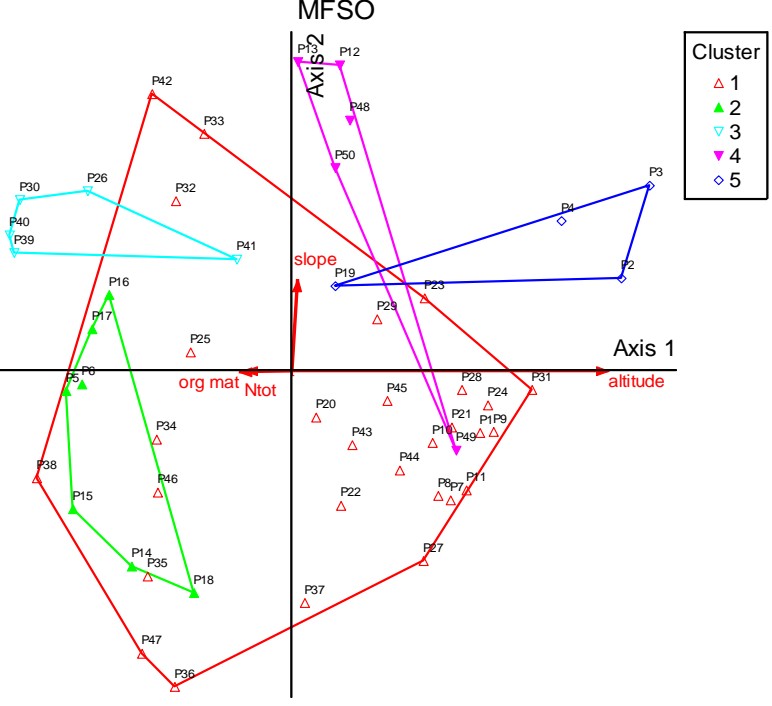

**Figure 4.** Biplot of multivariate fuzzy set ordination superponed with cluster analysis, the x-axis biplex represents the height gradient and the y-gradient represent nutritient gradient.

Indicator Species Analysis (ISA) shows indicator species for individual clusters (IV = indicator value, *p* = probability; A, B, C = layers in stand structure). The basis for the ISA was vegetation relevés made for each researched experimental plot.

Indicator species for **cluster 1**: Acer campestreC (IV = 51.7, *p* = 0.005), Clematis vitalbaB (IV = 38.8, *p* = 0.033), Cornus mas (IV = 44.6, *p* = 0.014).

Indicator species for **cluster 2**: Asparagus acutifolius (IV = 41.1, *p* = 0.017), Brachypodium sylvaticum (IV = 38.3, *p* = 0.04), Dactylis hispanica (IV = 44.3, *p* = 0.014), Phyllirea latifolia (IV = 28.6, *p* = 0.032), Quercus ilexC (IV = 42.9, *p* = 0.012), Rubia peregrina (IV = 28.6, *p* = 0.023), Rubus ulmifoliusB (IV = 50.0, 0.003), Salvia verbenacea (IV = 28.6, *p* = 0.034), Sonchus glaucescens (IV = 42.9, *p* = 0.014), Torilis japonica (IV = 42.9, *p* = 0.013).

Indicator species for **cluster 3**: Acer monspessulanumC (IV = 36.5, *p* = 0.026), Carpinus orientalisB (IV = 72.6, *p* = 0.002), Centaurea montana (IV = 50.0, *p* = 0.007), Chrysopogon gryllus (IV = 40.0, *p* = 0.023), Clematis flammula (IV = 47.9, *p* = 0.009), Cotinus cogyggria (IV = 49.6, *p* = 0.008), Festuca ovina (IV = 40.0, *p* = 0.023), Laurus nobilis (IV = 40.0, *p* = 0.025).

Ononis spinosa (IV = 40.0, *p* = 0.023), Quercus pubescensA (IV = 55.2, *p* = 0.001), Quercus pubescensC (IV = 36.2, 0.014), Rhamnus fallax (IV = 38.9, *p* = 0.023), Sorbus torminalisB (52.5, *p* = 0.006), Viburnum lantana (IV = 36.2, *p* = 0.034).

Indicator species for **cluster 4**: Acer obtusatumA2 (IV = 37.4, *p* = 0.028), Amelanchier ovalis (IV = 69.7, *p* = 0.001), Erica carnea (IV = 80.0, *p* = 0.001), Hepatica nobilis (IV = 36.8, *p* = 0.046), Molinia littoralis (IV = 40.0, *p* = 0.013), Ostrya carpinifoliaA (IV = 45.3, *p* = 0.015), Ostrya carpinifoliaB (IV = 43.3, *p* = 0.009), Phyteuma orbiculare (IV = 40.0, *p* = 0.013), Sesleria caerulea (IV = 40.0, 0.019), Sesleria tenuifolia (IV = 60.0, *p* = 0.001), Sorbus ariaA2 (IV = 52.3, *p* = 0.006).

Indicator species for **cluster 5**: Bromus erectus (IV = 49.1, *p* = 0.006), Bromus scoparius (IV = 28.6, *p* = 0.044), Chamaecytisus supinus (IV = 50.0, *p* = 0.005), Euphorbia seguieriana (IV = 50.0, *p* = 0.007), Galium sylvaticum (IV = 75.0, *p* = 0.002), Ligusticum mutellina (IV = 75.0, *p* = 0.002), Lonicera alpina (IV = 46.8, *p* = 0.005), Lonicera nigra (IV = 54.4, *p* = 0.007), Melica ciliata (IV = 59.7, *p* = 0.003), Moehringia muscosa (IV = 71.4, *p* = 0.002), Pinus nigraA (IV = 23.6, *p* = 0.002), Primula columnae (IV = 40.7, *p* = 0.01), Sorbus aucupariaB (IV = 40.7, 0.029), Thalictrum minus (IV = 61.0, *p* = 0.003), Valeriana tripteris (IV = 50.0, *p* = 0.005).

Figure 5 shows that the structure of the pine forest is irregular and has a diameter from 25.5 to 38.2 species per relevé. These data are related to different habitat conditions that determine the arrival of climatogenic vegetation.

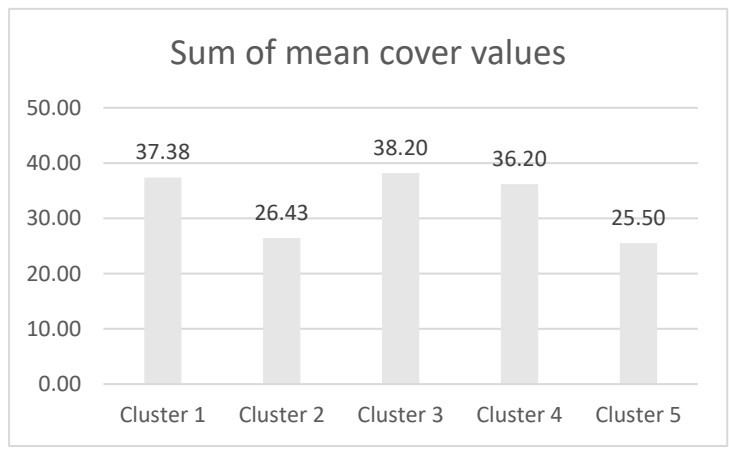

**Figure 5.** Shows sum of mean cover values for species of sub-Mediterranean plant species.

## 4. Discussion

Each habitat factor directly affects all the ecological factors upon which plant life is dependent, while the feedback effect of plants on the habitat is just as important. The influence of plants is most evident in the case of pioneer stands, such as forest plantations. Habitat factors such as climate (air temperature, humidity, precipitation, wind), soil (soil type, mechanical and chemical composition, structure), relief (elevation, slope, exposure) and biotic influences are strongly associated through the interrelationships

with ecological factors (light, heat, water, chemical and mechanical factors) [41,42]. The Mediterranean climate has two growing seasons, one in spring and the other in autumn, separated by a dry and hot summer period [43]. In recent decades we have combined effect of warming and drought [44]. The impact of climate change on vulnerability of Mediterranean-type biodiversity hotspots is likely to be highly dependent on the buffering effect of topographical complexity, and ergo, the existence and effectiveness of local refugia and microrefugia [45,46].

In the sub-Mediterranean habitats of Croatia, black pine primarily grows on limestone substrates (smaller ratio on dolomite), though it is also found on flysch substrates. It is most widely distributed in the sub-Mediterranean vegetation zone, where it is the most important species for reforestation of karst. This is also seen in other Mediterranean countries [47–49]. The habitat factors for black pine and other coniferous species in the Mediterranean, according to Quezel [50] and M'Hirit [51], were observed through the prism of the Emberger pluviometric coefficient. For black pine, it is significant that it appears in the humid, sub-humid and semi-arid zones.

In that sense, Trinajstić [19] stated that raising black pine plantations has phytogeographic justification only in the border areas between pubescent oak and beech areas, and that the closer these plantations to potential beech areas, the better the results will be. It was confirmed that black pine plantations, with regard to their composition and representation in the climazonal deciduous vegetation, is the best in the areas of pubescent oak forest, towards the borders of beech forests. From the aspect of amelioration, this is where these plantations can have the most direct and positive influence on the habitat conditions. Furthermore, in variable habitat conditions, as only forest plantations and natural stands older than 40 years were examined, there were no records, or only limited and individual findings of sprouting and saplings of black pine in a small number of plots. In the statistical analyses that divided the experimental plots into five main clusters, the largest cluster (29 plots) were situated in such vegetation areas as described above. Taking into account the productivity and economic role, in this cluster there are seven plots (P22, P27, P28, P29, P31, P32, P33) at elevations from 440 m (P22) to 714 m (P27) that stand out in particular. The edaphic conditions in these plots are favourable, both for the regeneration of climazonal vegetation, and for possible raising of new black pine plantations. The assessments depend on the manner and objectives of forest management. For example, silvicultural treatments, normally used for ecological or economic purposes, could have positive effect on biodiversity and biological interactions [48].

The natural stands of black pine (*Euphorbio triflorae-Pinetum nigrae* Trinajstić 1999) and (*Cotoneastro-Pinetum nigrae* Ht. 1938) are grouped into the fourth and fifth cluster, and are all at higher elevations (990 to 1082 m). The common ecological characteristics of these plots is the exceptionally strong influence of the dominant, cold, north-easterly Bura wind. In such habitat conditions, black pine has no competition from other species in the tree layer. An analysis of species indicators showed black pine as the indicator species only in the fifth cluster. In the other clusters, most species were characteristic for the sub-Mediterranean zone.

The edaphic factor is certainly decisive in the return of climazonal vegetation, particularly the limited soil horizon, i.e., forest floor. Our results showed (Figure 4 and Table 2) that nutrient gradient has a significantly lower value. We believe that this is to be expected, because it is mainly due to relatively shallow soils, and pines cannot have a positive amelioration effect during one rotation period (80 years). These two gradients of reliefs have the greatest impact on the structure of vegetation in the footage. The feed gradient has a much smaller effect as seen by the length of the vector. This means that the nutrient accumulation, ie the soil melioration effect on the total sample, may have a very limited effect and is generally not significant. It can only be effective in some places that can be determined by analyzing individual clusters.

In the study area, black pine stands are in the landscape with active erosion processes associated with soil degradation and the reduction in valuable land, as in other Mediter-

ranean karst areas [52]. According to Martinović [18], the forest floor is a natural organic fertilizer containing all the nutritional elements required by forest tree species. The forest floor is important for regulating the hydrological conditions of the soil, reducing evaporation and increasing infiltration of precipitation, while it also positively affects temperature relations within the soil. It is closely associated with the general condition of humus in the soil, and its composition and properties depend on numerous factors and processes. One of the elementary processes is humus formation.

The influence of black pine is also considered in the relationship of soil types and productivity [18]. This leads to the production order of soils and specific indicators of soil fertility. There is stronger humus formation in black pine plantation in the northern Adriatic area than under natural forest vegetation, and that the surface layer of soil is richer in total nitrogen and physiologically active potassium. From the research results, and also in relation to the study by Trinajstić [19], in the phytogeographic sense of the productivity of black pine stands, it can be concluded that black pine plantations have their highest productivity on organic mineral black soils (higher sub-Mediterranean belt), followed by brown soil on limestone, and then on rendzina (lower sub-Mediterranean belt). The black pine forests in the Mediterranean have equal commercial and ecological importance, and therefore have been included in the list of Europe's threatened habitats [27]. However, a problem arises in the lack of successful regeneration of those forests [14], which is also seen in the karst areas of Croatia. As some author mentioned natural regeneration is a main phase of forest dynamics and the only way to guarantee forest persistence in the long term [53]. In the relationship between soil and vegetation on karst, it must be considered how Mediterranean forest ecosystems cannot be considered for their production function alone; however, if these are commercial forests, they must meet the objectives of forest management. In the study area, the advantage is that most of the black pine forest plantation (included the cluster with 29 plots) are from pioneer stands, and near the end of their rotation period, shift in the direction of stand conversion. The process of progressive succession can be seen, with the regeneration of the sub-Mediterranean climazonal deciduous vegetation. Similar results can be seen in Tesei et al. [54], where the structure of the pine forest is irregular and has an average of 29.5 species per relevé (similar to our research, Figure 5). The dominant tree layer is monospecific as *Pinus nigra*, with an average height of 20 m and mean coverage of 70% to 80% and the shrub layer was well represented. The conversion process of forest cultures and conifer plantations is considered by many authors to be both significant and necessary [55–57]. Afforestation efforts to enhance adaptation of these forests in future climate scenarios will then require the selection of more drought-resistant genotypes and species and transformation to mixed, multi-aged forest stands [58]. This process may accelerate silvicultural works and in earlier thinning years to increase the quality of wood in the stand. Unfortunately, these works were not conducted in the study area during the prescribed 80-year harvest cycle (rotation period). In line with the forest management plan, as the document that governs the management of the forests and forest lands, 2000–2500 samplings of black pine per hectare are envisaged for raising a forest plantation. The first thinning cycles are planned between 20 and 30 years, and already at that time the number of trees should be reduced to 1300 to 1500 trees per hectare. Thinning should be continued every 10 years until the end of the rotation period (80 years). In comparing this with the research results, it is evident that there are deviations of wood volume on most of the experimental plots. The main reason for this is the lack of silvicultural works in the stands (cleaning and thinning). These works were only conducted in a small number of forest plantations as weak thinning campaigns and sanitary cutting due to fires in open spaces. Similar works in black pine plantations was also recorded by Zlatanov et al. [59]. Given that these results indicate the underperformance of wood volume, different strategies are need for the management of these forests. Some authors [59] have grouped the silvicultural strategies for the conversion of coniferous plantations. In line with these recommendations, the experimental plots could be grouped according to: (1) the capability for stand conversion after one harvest

cycle of black pine, or raising new forest cultures on the same habitat (epi-Mediterranean or higher sections of the sub-Mediterranean zone); (2) stand conversion is possible, though gradually, and with limited and smaller works aimed at obtaining a mixed deciduous and coniferous forest, with later support of succession towards the climazonal vegetation (sub-Mediterranean zone); and (3) conversion after one rotation period is not possible, and at least one more rotation period of black pine is needed, the growth and development of black pine is slowed due to the unfavourable habitat conditions and higher slopes (lower sections of sub-Mediterranean zone towards the eu-Mediterranean zone). Most of the current issues and hindering conditions for conversion are tied to the lack of any silviculture works (cleaning) of forest plantations in the early years and thinning in later years of the pine forest plantation.

## 5. Conclusions

The sub-Mediterranean habitats in western Croatia are recognisable for two species. Pubescent oak (*Quercus pubescens* L.) is the climatogenic deciduous species and black pine (*Pinus nigra* J.F. Arnold) is the main forestation species of this area. The differences in the synecological factors in the research area resulted in the presence of sub-Mediterranean, epi-Mediterranean and eu-Mediterranean elements within the black pine forest plantations. With increasing elevation in zones with higher precipitation, the ecological conditions were better, and the production potential was also more pronounced in those habitats. In the study area, this production potential needs to be improved and better utilised. Forest plantations in which silvicultural works are periodically carried out are better and stronger in the qualitative sense. This also reduces the risk of the occurrence and development of large wildfires. The lack of these works reduces the return of the indigenous deciduous vegetation, while also creating greater quantities of forest fuel (dry lower branches, dry standing trees due to density). These facts should be considered when aiming to achieve higher quality forest plantations of black pine. The research results and previous experience indicate that within the epi-Mediterranean zone of Croatia, new forest cultures of black pine can be raised (after one rotation period), as they can fulfil both the desired ecological and commercial roles. In the sub-Mediterranean zone of Croatia, the conditions are less favourable (edaphic conditions, higher slopes, strong influence of wind), and therefore greater productivity of black pine is lacking these areas.

**Author Contributions:** Conceptualization, D.B.; methodology, D.B.; software, V.H.; validation, V.H. and D.B.; formal analysis, V.H. and D.B.; investigation, D.B., V.H. and M.A.; resources, D.B. and R.R.; data curation, D.B.; writing—original draft preparation, D.B.; writing—review and editing, D.B. and V.H.; visualization, M.A. and R.R.; supervision, V.H.; project administration, D.B.; funding acquisition, R.R. All authors have read and agreed to the published version of the manuscript.

**Funding:** This research received no external funding.

**Data Availability Statement:** Not applicable.

**Acknowledgments:** The authors thank the employees of the public enterprise „Hrvatske šume" d.o.o. in the field of research of research (Croatian Forests). This study was supported by human and material resources of public enterprise „Hrvatske šume" d.o.o.

**Conflicts of Interest:** The authors declare no conflict of interest.

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
