# Peer review of "Ecological Potential of Mediterranean Habitats for Black Pine (Pinus nigra J.F. Arnold) in Croatia"

_forests, doi:10.3390/f13111900_

Round 1
Reviewer 1 Report
A brief summary:
The scope of the article, to determine the habitat factors that have the most important influence on the black pine both in natural stands and in plantations is clear and brings an important contribution and an interesting approach to the problem of "what type of vegetation to be promoted in Croatian Mediterranean habitats in the present-day local and general climate-change context".
The article text is well-structured, and relevant to the field of forests and forestry. The key concepts presented in the article are well-defined.
The cited references are relevant.
The article is sound and benefits from an appropriate methodology in line with the objectives as well as from a broad and adequate experimental design and statistical analysis. Figures and images are generally appropriate. Results are reproducible and conclusions are relevant to the article's objectives.
General concept comments:
Better compliance between the title "Ecological potential" and the objective of the study: "to determine the habitat factors that have the greatest impact on the growth and development of black pine..." only, would be beneficial.
The scope and objectives of the study should be more clearly presented. I.e. A more detailed statement about "differences" to be determined as objectives between "...black pine and black pine forest cultures .... and ...reforestation with black pine".
Compliance with Natura 2000 objectives as well as the black pine natural regeneration is very little considered in the study area.
The study is about Mediterranean habitats but it is limited to Croatia area. A more extensive synthesis of recent articles on black pine (as a species with endemic taxonomic subunits) vulnerability and behavior during the very strong and prolonged drought in the recent period in various Mediterranean habitats could bring more consistency to the introduction as well as to the discussions (i.e. Bayar et Deligoz, 2021; Tischer et al, 2018; Navarro Cerrillo et al, 2018...).
The experiment is well-designed to cover the research aim and objectives and the research methodology is appropriate but more details are required about the ..." initial preliminary analysis made to determine the number and order of the variables involved".
The figures, schemes, and tables are in general appropriate but some minor changes or additional details could make them more clear and the data easier to understand and interpret (i.e. those describing the results from Indicator Species Analysis or results of Multivariate Fuzzy Set Ordination). Some data are presented as results (table 1) but they include historical data.
The study results should be more clearly discussed and compared with other similar research results in Discussions. The Discussion text is too descriptive, a part of it should be moved to the Introduction.
It looks that there is some inconsistency regarding the role of soil/nutrient accumulation between the results and conclusions, which is also reflected in the abstract text.
Specific comments:
-There is a repetition in the text (line 11-13 is similar to line 15-17).
- line 153 - explanation and details are missing about how .."....an initial preliminary analysis was made to determine the number and order of variables involved..." (line 153-154).
-table 1 (line 162) is presented as results but contains some historical data (annual precipitation, annual temperature) without indicating the data source.
-Fig. 4 (line 196) and table 2 (line 200) need full details of the elements described.
-Line 208: Some words are missing: "...and the y gradient ...".
-The text included between Lines 214-242 can be better integrated into a new table.
-Fig. 5 elements need a clear and detailed explanation. The title text of fig. 5 must be reviewed.
-the line 210-213:" ie The soil melioration effect on the total sample, may have a very limited effect" (Results) is not commented in line with the text from lines 292-294 (Discussion): "The edaphic factor is certainly decisive in return of climazonal vegetation,..."
-Line 293-294, the sentence is not complete: "...Landscape with active erosion processes, associated with soil degradation and the reduction of valuable land"[48].
-line 321... and line 353: "conversation"...must be corrected to ..."conversion"...
- conclusions are in general in line with the obtained results but line 368: ..." In the study area this protection potential needs to be improved and better utilized" is the right conclusion but it is not substantiated in the article results and discussions.
Regarding the role of soil types/nutrient accumulation, some inconsistency is present in the text between abstract (line 17-19), results (line 210-212) and discussions (line 292) or conclusions (line 378).
Author Response
Please see the attachement

Reviewer 2 Report
In the manuscript titled “Ecological potential of Mediterranean habitats for black pine (Pinus nigra J.F. Arnold) in Croatia” the authors report the habitat factors that have the greatest impact on the growth and development of black pine plantations and natural black pine forest and on the return of climazonal deciduous vegetation. I think that the objective is certainly very interesting. The only major comment is that the description of the results need a little more detail.
English language and grammar in manuscript need improvement.
Please see below comments and suggestions:
Keywords: replace “pine culture” with “pine plantations”
Materials and Methods: Study area describes locations of the study, but does not include any information on climate or soils. I suggest providing more information on items such as soils, average annual rainfall and temperature. It is not clear in the methods which environmental variables do you used in the analyses. Please define the variables. Results: This part is not clear enough, should be better developed. The possible importance or implications of the results is not enough transmitted. Please, provide a concise and precise description of all your Tables and Figures shown in the manuscript. In Results and Discussion part, key messages should be provided. Figures: it is necessary to correct the Figure title in section Results - Figure should have a short explanatory title and caption to help the reader to understand what you have shown in the Figure.
Line 10: replace the word “cultures” with “plantations”
Line 11: replace the word “stands” with “forests”
Line: 11-14. correct as indicated: “The objective of the study ….development of natural black pine forests and black pine forest plantations on the return …. black pine.”
Line 14: replace the word “isolated” with “selected”
Line 15: replace the word “cultures” with “plantations”. And correct/replace it further in the text
Line 15-17: delete sentence “The objective of the study was to determine 15 the habitat factors that have the greatest impact on the growth and development of black pine and 16 black pine cultures on the return of climazonal deciduous vegetation.”, it is already written in the sentence before (line 11-14)
Line 25: replace keyword “pine culture” with “pine plantations”
Line 58: delete “this” in the beginning of the sentence. The sentence begins with “In addition to…”
Line 71-84: unclear paragraph
Line 83-84: correct the sentence “Therefore, Santana [9] warns of the key ...”
Line 86-88: correct the sentence “Namely, habitat conditions (slope, shallow soil) and, fire largely governs the management …”
Line 99-104: indicate Figure 1 and 2 in the text
Line 113: the sentence “All plots were set up in stands over 40 years old.” move in the end of the section: Study area, (line 104)
Line 113-114: unclear sentence “For each plot, the girth of black pine trees was measured by girth classes, number of trees, basal area and wood volume.” How girth can be measured by girth classes?
Line 116-117: Please delete “over an area of 25 x 25 m (625 m2)”. It was already written, in the beginning of the paragraph.
Line 123-127: delete sentences “Soil reaction is a significant ... processes in the soil can be determined.” this is not relevant for section Materials and Methods.
Line 153: list the environmental variables that were taken for analysis
Line 161: Results section have to begin with text, not with table
Line: Table1. Heading of column have to be corrected - word have to be in one line (e.g. Phytocenosis); widen the column. Table 1. delete from results section and put it in M&M-study area
Line 164-173: Correct the paragraph, it is unclear;
line 164-166: part “The bioclimatically explored area ……. winds dominate.” and Table 1- move to section Materials and Methods–study area.
Line 166: before sentence “The first cluster…” explain what the cluster analysis present. Suggestion: “According to the cluster analysis, the obtained optimal number of clusters was 5. Figure 2 represents ...
Line 207-208: “The x-axis biplex represents the height gradient and the y-gradient.” Unclear sentence
Line 208-213: “These two gradients of reliefs have the greatest impact on………. some places that can be determined by analyzing individual clusters.” move to Discussion section
Line 242: Discuss Figure 5 in the section Results; Fig5 is presented but not discussed
Line 253-256: delete sentences “The Mediterranean climate has two growing seasons ….effect of warming and drought.”. They are not relevant for the context of paragraph. Or reformulate the sentences.
Line 269: insert “the” before word “best”
Line 293-294: unclear: “Landscape with active erosion processes, associated with soil degradation and the reduction of valuable land.”
Line 353: word “conversation” replace with “conversion”
Author Response
Please see the attachement
